# Gridded Visibility Products over Marine Environments Based on Artificial Neural Network Analysis

**Yulong Shan** [1] **, Ren Zhang** [1,*]**, Ismail Gultepe** [2,3]**, Yaojia Zhang** [4]**, Ming Li** [1] **and Yangjun Wang** [1]

1 School of National University of Defense Technology, Nanjing 210000, China; oceanpapersyl@163.com (Y.S.); mingli152@163.com (M.L.); wyjscholar@sina.com (Y.W.)
2 MRD, Environment and Climate Change Canada, Toronto, ON M3H 5T4, Canada; ismail.gultepe@canada.ca
3 Faculty of Engineering and Applied Science, UOIT, Oshawa, ON M3H 5T4, Canada
4 Business School, University of Birmingham, Birmingham B15 2TT, UK; YJZpaper@163.com
* Correspondence: Zrpaper@sohu.com; Tel.: +86-13260786710

**Abstract:** The reconstruction and monitoring of visibility over marine environments is critically important because of a lack of observations. To travel safely in marine environments, a high quality of visibility data is needed to evaluate navigation risk. Currently, although visibility is available through numerical weather prediction models as well as ground and spaceborne remote sensing platforms and ship measurements, issues still exist over the remote marine environments and northern latitudes. To improve visibility prediction and reduce navigational risks, gridded visibility data based on artificial neural network analysis can be used over marine environments, and the problem can be regarded as an air quality prediction problem based on machine learning algorithms. This new method based on artificial intelligence techniques developed here is tested over the Indian Ocean. The mean error of the inferred visibility from the artificial neural network analysis is found to be less than 8.0%. The results suggested that satellite-based optical thickness and numerical model-based reanalysis data can be used to infer gridded visibility values based on artificial neural network analysis, and that could help us reconstruct and monitor surface gridded visibility values over marine and remote environments.

**Keywords:** marine visibility; aerosol optical depth; artificial neural network

## 1. Introduction

With the warming of the Earth's surface and the gradually opening of the Arctic passages [1], marine transportation has become much more important than before. Since the natural environment is complex and harsh in some sea areas, an objective assessment of the environmental risks of crossing the marine areas is needed urgently. The main factors that affect the navigation risk are sea wave, sea ice, visibility (Vis), icing, wind, and turbulence, as well as precipitation [2,3]. Information on Vis and freezing precipitation over marine environments are usually limited and not easy to be predicted or monitored over various scales [3,4]. For this reason, having gridded data of Vis with high quality is a precondition to assess the navigation risk in marine environments. At the same time, Vis has an important influence on aviation, especially on the take-off and landing for pilots. Due to that, Vis is a direct reflection of air quality; the gridded Vis data could help people know more about the air quality around the world and monitor and predict the air quality.

The statistical data on marine accidents [5] showed that 220 maritime accidents occurred in the southern waters of the Taiwan Strait from 2000 to 2007, including 122 collisions, accounting for

55.45% of the total number of maritime accidents (68 collisions) that occurred during poor visibility conditions. At the same time, in China, 31% of the total number of aviation accidents were caused by adverse weather. Even in the United States, where aviation technology is advanced, weather-related air crashes account for up to one-third of all fatalities [6]. Therefore, in aeronautical meteorology, people have always attached great importance to the relationship between Vis and aviation. These suggest that under the Arctic environments, these numbers can easily go beyond what is given here because of the lack of observations near coastal areas and open marine environments. In fact, because of the lack of observations, including ones from satellites over Arctic regions, and limited precipitation radars available on the large ships (not on small ones), fog radars (e.g., W-band radars) are needed to monitor fog conditions that are not usually available. Accurate nowcasting and forecasting of Vis is needed more than using only monitoring of the ship maneuvers and their motion. This is in a way similar to aviation applications [7]. The only way to predict fog currently in the Arctic is use numerical weather prediction (NWP) models, but their physical algorithms are usually not developed for fog predictions [8].

Vis is the greatest horizontal distance at which it is possible to observe and identify particular objects, and fog is defined when Vis <1 km. For Vis evaluation, any particle type in the atmosphere can play an important role and needs to be considered in the analysis [7]. Currently, Vis data are available from in-situ and remote sensing platforms, as well as from NWP simulations in a limited way. In in-situ observation for Vis, the atmospheric extinction coefficient is usually measured firstly based on an instrument. Then, the Vis value is calculated based on the Koschmieder equation, which defines visibility mathematically. Many methods have been proposed for remote sensing observations on Vis. The most often used methods to obtain Vis data based on satellite remote sensing is to obtain the planet reflectivity at various bands firstly, and then calculate the Vis data based on the built relationship between Vis and planet reflectivity. In recent years, one way to calculate Vis based on satellite-based aerosol products has been proposed, which has been tested on the East Coast of the United States [9]. However, Vis from satellite observations cannot be used accurately during the daytime when mid or high-level clouds exist [8,10,11]. At the same time, Vis data from the NWPs is limited because of the high uncertainty in the Vis microphysical parameterization based on using only liquid water content (LWC) [10,12], where droplet number concentration (Nd) is assumed to be a negligible factor. In addition, because of the limited number of observations from buoys and ships, the reliable Vis data over marine environments can be highly uncertain; therefore, both the monitoring and forecasting of Vis need to be improved significantly.

Many earlier studies have tried to obtain Vis data over gridded surfaces using observations and NWP models. For example, Fei et al. [13] retrieved Vis based on principal component regression (PCR) and NOAA/AVHRR remote sensing data. They found that the correlation coefficient between the retrieved Vis and measured Vis was 0.82, but the method could not accurately obtain high or low Vis, and the average relative error was at 21.4%. A new microphysical parameterization for fog Vis (MPF) method using RH (relative humidity), LWC, and Nd parameters was developed by Gultepe et al. [10]. They stated that when the RH is close to 100%, the MPF should be used in NWP simulations. This new method significantly improves the prediction of Vis obtained from the operational forecast models. Uncertainty in the new Vis parameterization is found to be less than 29% compared to more than 50% based on only LWC-based parameterization. Their work improved NWP predictions of Vis significantly, but further studies are still needed for high-resolution prediction models. Kessner et al. [9] studied the relationship between Vis and satellite-based aerosol products such as optical thickness (aerosol optical depth, or AOD), and presented their results for atmospheric attenuation coefficients over the globe.

The fog prediction of Vis in marine environments using NWP models includes large uncertainties on small space scales over the short time periods, because the microphysical schemes used in numerical models [4] have limited boundary layer processes such as turbulence. To achieve more stable and reliable historical gridded Vis data around the world's marine environments, remote sensing retrievals of

aerosol and cloud optical thickness together with meteorological parameters are needed. There have been many earlier studies revealing the relationships between atmospheric Vis and meteorological elements. For example, Hong [14] studied the various environmental factors affecting atmospheric Vis in the region of Chongqing, China, and their results showed that atmospheric Vis changes seasonally and also as a function of horizontal wind speed (Uh), RH, and rainfall. Founda et al. [15] explored the interdecadal variability and trends of surface horizontal Vis in the urban areas of Athens. Their results showed that Vis was negatively correlated with RH. The correlation was stronger in the early part of the time series, and decreased over the years. In contrast, a positive correlation between Vis and wind speed was found. At the same time, the relationship between AOD and Vis in Athens was also examined in the study, and their negative correlation was confirmed. Founda et al. also studied the air quality of Athens by studying the Vis in the area. Singh et al. [16] investigated the long-term trends in Vis for eight meteorological stations situated in the UK. In general, Vis has improved at most of the stations through time, which are attributed to reductions in aerosol particle loadings and decreases in RH. At the same time, variations of a few percentage points in this RH range can have significant effects on Vis. Their study also showed that Uh and wind direction were both important factors influencing Vis. Gultepe et al. [7] summarized all the meteorological factors that affect Vis severely.

Since precipitation and fog both have a great effect on Vis, meteorological processes affecting precipitation and sea fog should also be considered in the analysis when inferring Vis from remote sensing platforms and NWP model simulations. Zhao et al. [17] studied the relationship between pressure field and rainfall, and stated that the pressure field was significantly correlated with weather systems producing rain. This is likely due to a short-wave trough with large temperature gradients. Low fog visibilities over the ocean are also found to be strongly related to differences between sea surface temperature (Ts) and air temperature (Ta). Qu et al. [18] also studied the formation of sea fog in the Bohai Sea, and emphasized that the temperature difference between ocean surface and air temperature (Ts-Ta) (TDsa) played an important role on the formation of sea fog type and occurrence. These works suggest that meteorological parameters such as LWC, Nd, RH, Uh, Ta, Ts, and pressure should be considered in Vis calculations [7], where LWC is defined as the quality of liquid water in a unit volume of air, Nd is defined as number of droplets in a unit volume of air, and RH is defined as the ratio of absolute humidity in the air to the saturated absolute humidity at the same temperature and pressure.

Considering that a relationship between Vis and both atmospheric optical thickness and meteorological parameters exists, a non-linear fitting method can be used to obtain a relationship between Vis and other parameters. The artificial neural network (ANN) method used in the analysis here utilized the satellite-based aerosol/cloud products plus reanalyzed data from the European Centre for Medium-Range Weather Forecasts (ECMWF) to infer gridded Vis, and generated Vis over the Indian Ocean to test the accuracy of the model.

Overall, the main objective of this work, using in situ and remote sensing observations and reanalysis data, is to generate an accurate and reliable gridded horizontal Vis data set that in the past around the world was used for mainly marine environments based on an ANN analysis. Section 2 provides info on the data and ANN model description. Section 3 provides the results and discussion. Then, Section 4 provides the conclusions.

## 2. Data and ANN Model

In data processing, Vis changes related to meteorological parameters are studied first (Section 1), and then relationships are set up using ANN. After finding out which meteorological factors affect Vis, the factors are elected to be used in the training of ANN to infer Vis. During this training, we eliminated insignificant meteorological parameters by making its weight influencing Vis small. Note that Vis here represents not only that for fog and precipitation, but also that for aerosols. Then, the ANN model is built on the selected factors, and the ANN model is trained with measured Vis and met

parameters data, as well as with AOD. During analysis, Vis representing fog and precipitation was inferred using reanalysis gridded data from ECMWF and AOD from the MODIS (Moderate Resolution Imaging Spectrometer) satellite. Vis for both precipitation and fog was inferred using only reanalysis gridded data from ECMWF if satellite-based AOD was unavailable when the clouds were thick enough above the boundary layer. In the final step, inferred Vis was then corrected using measurements. The technical flowchart to infer gridded Vis is given in Figure 1.

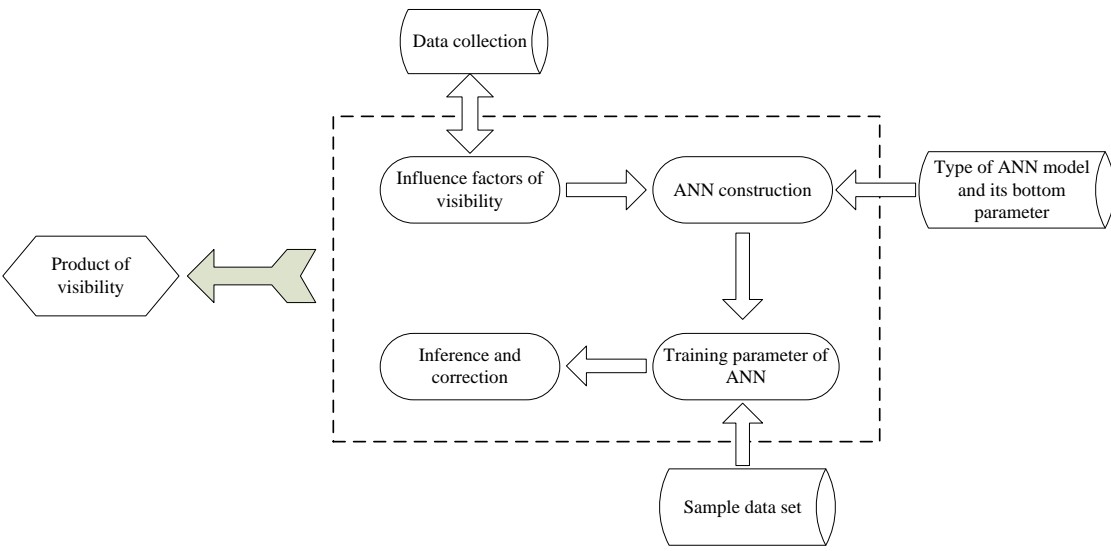

**Figure 1.** The technical flowchart to infer visibility (Vis) based on artificial neural network (ANN).

## 2.1. Data

In the analysis, AOD, SLP (sea level pressure), Ta at the 10 m, RH, Ts, TDsa, and Uh are considered important parameters affecting Vis. These parameters affecting Vis as a physical/optical process are shown as follows:

AOD: AOD is defined as the integration of extinction coefficients in the vertical direction, which describes the reduction effect of aerosols for light.

SLP: SLP has an important effect on precipitation, and precipitation has a great effect on Vis.

Ta: Ta has an important influence on the particle size distribution and gas–solid distribution of air pollutants. The higher the temperature near the surface, the stronger the convection, the lower the concentration of pollutants, and the better the Vis

RH: RH is not only closely related to precipitation, but also affects the concentration of aerosol chemical components, which is closely related to the scattering coefficient of particulate matter.

Ts and TDsa: On the one hand, the higher Ts could cause the stronger convection near the surface, which lead the lower concentration of pollutants and the better Vis. On the other hand, Ts and TDsa are closely related to the occurrence of sea fog [18].

Uh: Uh cannot only dilute atmospheric pollutants, but also change atmospheric stability and has a greater impact on Vis. Usually, a greater Uh could cause better Vis.

The data of meteorological parameters used to train ANN are taken from the International Comprehensive Ocean-Atmosphere Data Set (ICOADS), which is the largest collection of ocean surface observational data sets covering the period from 1784 to the present, including the data of the ships, buoys, and coastal sites from all parts of the world. Data of AOD are taken from MODIS Level 3 gridded atmosphere daily global joint product at a spatial resolution of 1° × 1°, and that includes all aerosol types [9]. The data of gridded meteorological parameters are used to infer the historical gridded Vis data obtained from ECMWF model runs, and tropical cyclone data that were used to further test the accuracy of inferred Vis were obtained from the Joint Typhoon Warning Centre (JTWC).

Due to the inconsistent position of AOD data and ICOADS data in the areas studied, it was necessary to interpolate ICOADS and AOD data to the same position using a statistical method. Then, we were able to get the complete sample data set to train the ANN model. The statistical method used to govern this extrapolation in the study is the Inverse Distance Weighting Method [19], which assumed that AOD at the position of ICOADS data could be influenced by the AODs within a radius of R km. If there are Q numbers of locations in which AOD data are located within an R-km radius near the position of ICOADS data, then the AOD at the position of ICOADS data can be defined as:

$$\text{AOD} = \sum_{k=1}^{k=Q} aod_k * \lambda_k \tag{1}$$

where $aod_k$ is the AOD value at the Kth location, $\lambda_k$ is the weight of the AOD value at the Kth location influencing the AOD at the position of the ICOADS data. $\lambda_k$ can be defined as:

$$\lambda_k = \frac{1/D_k}{\sum_{k=1}^{k=Q}[1/D_k]} \tag{2}$$

where $D_k$ is the distance between the location $k$ and the position of the ICOADS data located.

Since the radius R is closely related to the AOD at the position of ICOADS data located, it is necessary to find the R values making the generated AOD have an optimum value relative to the measured AOD. To do this, we first randomly selected 10% of the data set to test the accuracy of the generated AOD, and the remaining 90% were used to generate the AOD in the location of the randomly selected data. The initial radius was set to 20 km, the radius difference between two iterations was set to 10 km, and then, the mean relative errors of the generated AOD with different R values were calculated. Using AOD data from 1 January 2010, we studied the variation of relative error with R and an upper limit of the radius was set up as 500 km. Note that the AOD used to find R was from a MODIS level 2 product, which has a spatial resolution of 10 × 10 km. Note that MODIS level 2 products have a higher spatial resolution than level 3 products, and that can help to better study the effect of interpolation changing with R. The flowchart to find the best R is given (Figure 2), and the code for this is shown in the supplementary documents.

## 2.2. ANN Model

Artificial neural networks have been studied since the 1940s, and have been widely applied in the military, medical science, and broader economy, etc. This study employs a BP (back propagation) network to infer Vis data, which was first proposed by Rumelhart and McClelland in 1986 [20]. The BP network is one of the classical artificial neural networks, and can simulate any non-linear input–output relationship.

There are three layers in a BP network: the input layer, the hidden layer, and the output layer. The neurons are connected with others in the adjacent layers, but the neurons in the same layer are not connected with each other. The input and output layers are comprised of a single layer, but the hidden layer could have a different number of layers, and the number of neurons in each layer is also different depending upon the layer properties. We plot the topological structure of the BP network, which includes only one layer of the hidden layer (Figure 3).

From the structure chart, it is evident that the signal is transmitted from the input layer to the output layer. The output value of each node in the hidden layer depends on the input value, the weight of each node in the input layer influencing the node value in the hidden layer, and the threshold value and activation function of each node in the hidden layer.

$$X'_j = f\left(\sum_{i=1}^{n}\left(\omega_{ij} * X_i\right) - \Theta_j\right) \tag{3}$$

where $X'_j$ is the output value of the *j*th neuron in the hidden layer when the input value was input, $\omega_{ij}$ is the weight where the ith value in the input layer influences the jth value in the hidden layer, $X_i$ is the ith value in the input vector, $\Theta_j$ is the threshold of the *j*th neuron in the hidden layer, and *f* is the activation function of the *j*th neuron in the hidden layer. The signal transmission rule between the input layer and hidden layer is the same as the transmission rule between the hidden layer and the output layer.

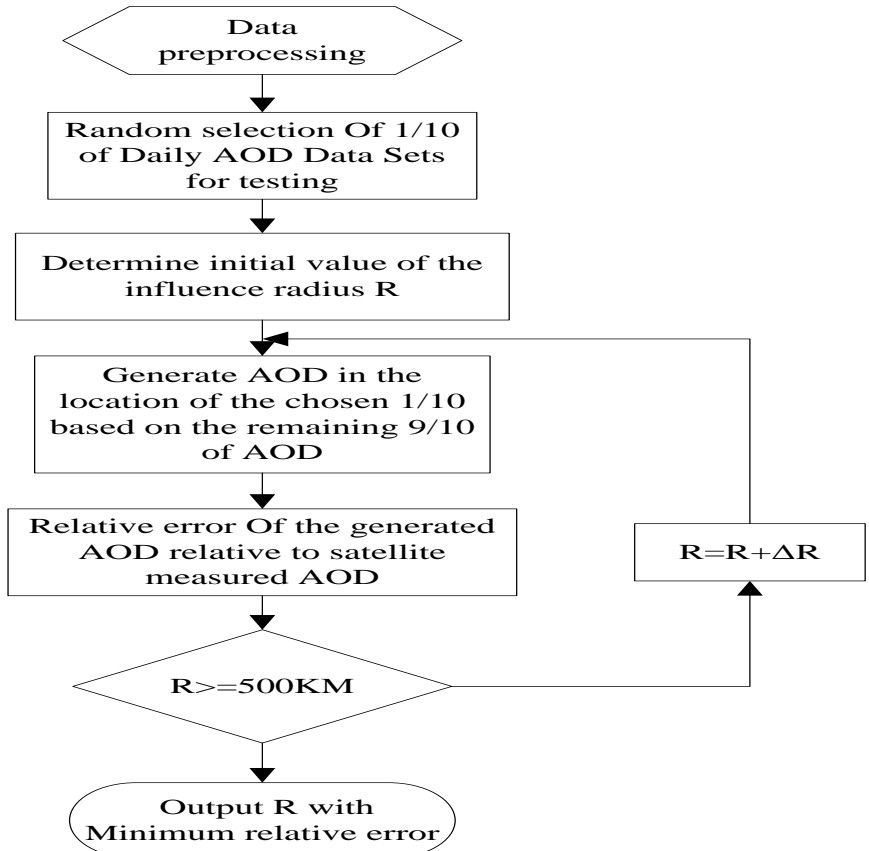

**Figure 2.** The technical flow chart to find the best R.

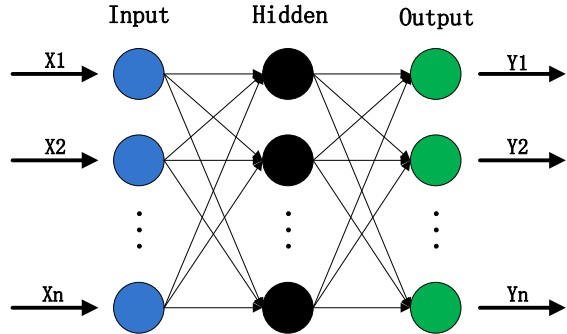

**Figure 3.** Topological structure of the back propagation (BP) network, which includes only one layer of the hidden layer.

The ANN model constructed in this study is trained based on a BP algorithm [21]. When there is an error between the actual output value and ideal output value, which form the output value in the training data set, the error would be transmitted from the output layer to the hidden layer and input layer to amend the weight and threshold of each neuron. The training process is complete only when the error is less than the convergence threshold. Artificial neural network technology is highly suitable for simulating the non-linear relationship between atmospheric visibility and the influencing factors.

This study constructs the BP neural network system with three layers based on the MATLAB toolbox. The number of nodes of hidden layers is considered to be 5, the maximum number of iterations is 100, the study rate is 0.1, and the convergence threshold is taken as 0.00001. Considering that Vis changes with monthly features [22], this paper inferred Vis for one month using ANN with the training data for that month. To test the fitting effect of the ANN model, the data from January 2010–2016 in the Indian Ocean (22.5° E–127.5° E, 70° S–25° N), representing a 7-year period, was employed to train the parameters of ANN, and the data from January 2017 was employed to test the parameters of the trained ANN model. To avoid the contingency of training results of neural networks due to the scarcity of training data, 30 experiments were repeated. Since the referred Vis grade based on ANN is not an integer, we defined its grade to the nearest grade. An average relative error of each observation point is defined as:

$$\delta = \frac{|V_c - V_r|}{V_r}.$$ (4)

where $\delta$ is the relative error, $V_c$ is the referred Vis, and $V_r$ is the measured Vis.

## 3. Results

### 3.1. Optimum Value Estimation

Since the contingency in the interpolation results of AOD always exists, 10 experiments were repeated. Figure 4 shows the average relative error of the generated AOD obtained from the 10 experiments with R from 10 km to 500 km. The analysis indicates that the relative error is the lowest when R is 20 km, and when R is between 20 km and 110 km, the relative error is below 15%. Due to the spatial resolution of used AOD being 10 × 10 km, we could get that the error of generated AOD increases with R. Since AOD data are always missing because of the thick clouds and the spatial resolution of the AOD used to infer Vis is 1° × 1°, this paper set R to 110 km to get more AOD data. Since the meteorological elements have a direct influence on sea fog and precipitation conditions and these two weather phenomena are the main factors affecting Vis, it's reasonable to ignore the effect of aerosols on Vis when inferring Vis, because large aerosols will only act as CCN (cloud condensation nuclei), leading to fog and cloud formation. So, if there is no AOD data within 110 km around the research location because of the thick clouds, which block the satellite's access to aerosol data, the effect of AOD on Vis is not considered in the analysis. However, if the clouds are high, then aerosols will play an important role in Vis.

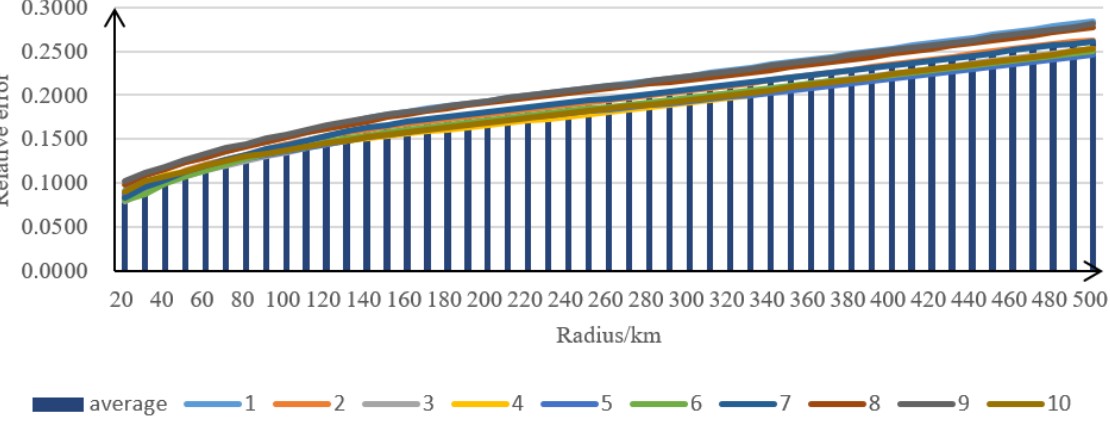

**Figure 4.** The average relative error of the generated aerosol optical depth (AOD) obtained from the 10 experiments with R values from 10 km to 500 km.

Vis is usually classified differently in application to navigation activities; this classification of Vis data in ICOADS is given at 10 levels (Table 1) [23]. When Vis mentioned in this study, it means a "visibility level" (see Table 1).

**Table 1.** The rule classifying the Vis level.

| Visibility Level | 1 | 2 | 3 | 4 | 5 | 6 | 7 | 8 | 9 | 10 |
|---|---|---|---|---|---|---|---|---|---|---|
| Visibility value (km) | <0.05 | 0.05–0.2 | 0.2–0.5 | 0.5–1 | 1–2 | 2–4 | 4–10 | 10–20 | 20–50 | >50 |

Since the transit time of the satellites monitoring AOD are 10:30 and 13:30 at local time, respectively, it is necessary to screen out data at this time from the ICOADS to make the data from ICOADS consistent with AOD in time. This paper assumed that AOD from 10:00 to 11:00 at local time is the same as AOD in 10:30 at local time, and AOD from 13:00 to 14:00 at local time is the same as AOD in 13:30 at local time, which is to get more data containing meteorological parameters and AOD. The sample data set to train the model for both fog and precipitation is shown in Table 2.

**Table 2.** The sample data set to train the model for both fog and precipitation.

| Sample Sequence | 1 | 2 | 3 | 4 | 5 | 6 | 7 | 8 | 9 |
|---|---|---|---|---|---|---|---|---|---|
| Year, month and day (UTC) | 20100101 | 20100101 | 20100101 | 20100101 | 20100101 | 20100101 | 20100101 | 20100101 | 20100101 |
| Hour (UTC) | 3:00 | 5:00 | 6:00 | 6:00 | 6:00 | 6:00 | 9:00 | 10:55 | 11:00 |
| Lon (°E) | 101.6 | 83.2 | 102.6 | 126.5 | 119.1 | 124.5 | 74 | 47.4 | 41 |
| Lat (°N) | −24.8 | 5.8 | −25.8 | 22 | 14.5 | −7.8 | 10.7 | 12.8 | 16.6 |
| Vis level | 8 | 8 | 8 | 9 | 9 | 9 | 8 | 1 | 9 |
| RH | 62 | 82.3 | 65.7 | 68.9 | 76.9 | 75.2 | 79.8 | 64.3 | 66.6 |
| Ts | 22.4 | 27 | 22.3 | 24 | 28 | 30 | 30.2 | 25.6 | 28 |
| Ta | 21.9 | 28 | 21.9 | 23 | 27.5 | 31 | 31 | 26.7 | 30 |
| Uh | 12.3 | 9.3 | 13.4 | 9.8 | 13.4 | 5.1 | 2.1 | 0.6 | 10.3 |
| SLP | 1018.6 | 1012 | 1017.1 | 1019 | 1007.4 | 1005.1 | 1011.1 | 1020.3 | 1012.2 |
| TDsa | 0.5 | −1 | 0.4 | 1 | 0.5 | −1 | −0.8 | −1.1 | −2 |
| AOD | 0.16 | 0.34 | 0.15 | 0.32 | 0.2 | 0.22 | 0.46 | 0.21 | 0.24 |

*3.2. Fitting Effect of ANN*

Since too much data needs to be shown in Figure 5 and one figure could not well show the result, we decided to use two figures to show the average relative error of each observation point between the inferred Vis and the measured Vis. The root mean square error (RMSE) of the inferred Vis is 0.9323 when considering AOD, and 0.9578 when neglecting AOD. Since the Vis data in this study is given at different levels, the RMSEs that are less than 1 should be regarded as small. Therefore, the ANN model constructed in this paper can highlight the relationship among Vis, AOD, and meteorological factors, whether or not AOD was taken into consideration, which also demonstrates the assumption above that it's reasonable to ignore the effect of aerosols on Vis when inferring Vis. At the same time, the error based on NWP and PCR can reach up to 29% and 21.4%, respectively [10,13,24], so the result based on ANN has a higher accuracy than the two common methods. However, there are a few inferred Vis with large relative error. Three reasons may cause this result. One of the reasons is the inadequate training of ANN due to the incomplete data set; the second reason is the deficiency of the model, such as the selected factors influencing Vis not being complete. The last reason is the inaccuracy of the data set used to test the accuracy of the trained model. All these three reasons combined likely played an important role for the accuracy of ANN analysis, which is discussed below. Note that the inferred Vis with a large relative error has a large relative error whether taking AOD into consideration or not, which indicates that the main reason for this is the first or the third reasons mentioned above.

Data used to further test the effect of ANN is from the Chinese Ninth Arctic Science Examination (CNASE), which began on 20 July and ended on 26 September, in 2018. The scientific expedition covered more than 12.5 million nautical miles, including 3815 nautical miles in the ice area. The northernmost

location of the expedition is 84.8° N. The data recorded in this expedition include meteorological and marine data. The meteorological data include air temperature, relative humidity, dew point temperature, sea-level pressure, wind speed, horizontal Vis, etc., which are recorded once a minute. The marine data include water temperature, salinity, etc., which are recorded once every 30 s.

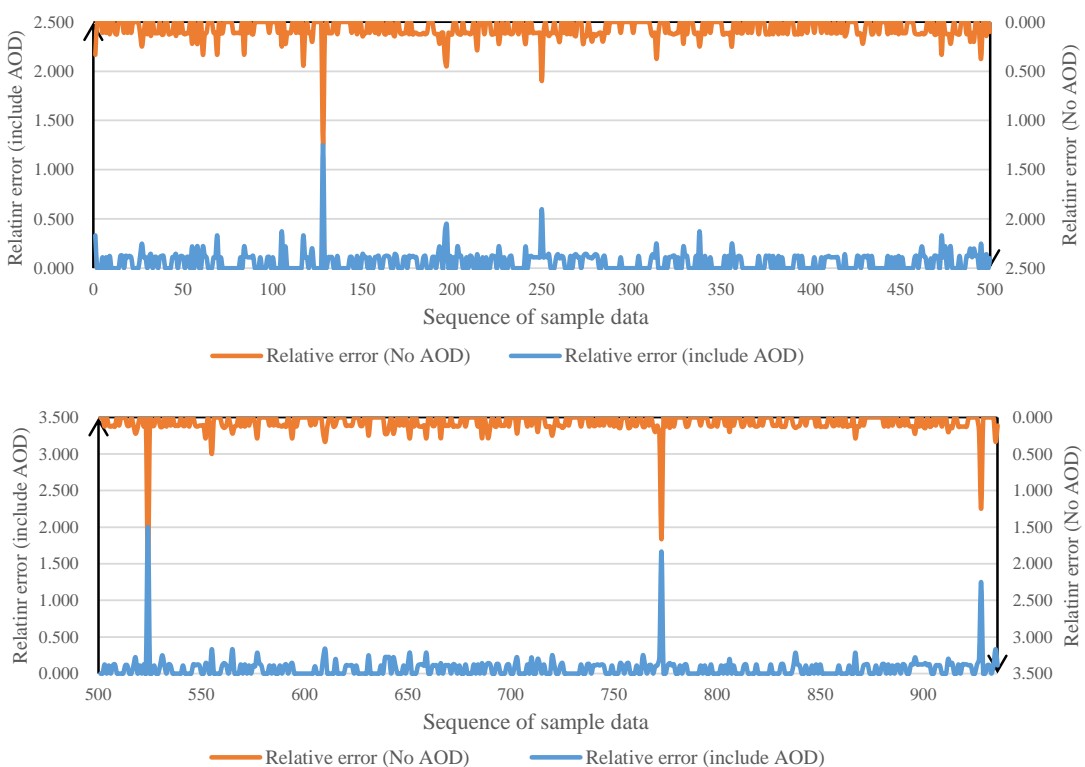

**Figure 5.** Composition of the average relative error of each observation point when taking AOD into consideration or not.

Data used to train the model is from ICOADS, from 2001 to 2015 in the Arctic (180° W–180° E, 60° N–90° N), and the data used to test the accuracy of the trained model is one-hour average data from CNASE. Since the sample data sets are different when inferring Vis in different months, this paper calculated the relative error in August and September, respectively. The inferred result is shown in Figure 6. The average relative error of all observations in August and September is 19.0% and 12.7%, respectively, which indicated that ANN could well fit the relationship between Vis and its influence factors. Note that the AOD was not considered when calculating Vis here.

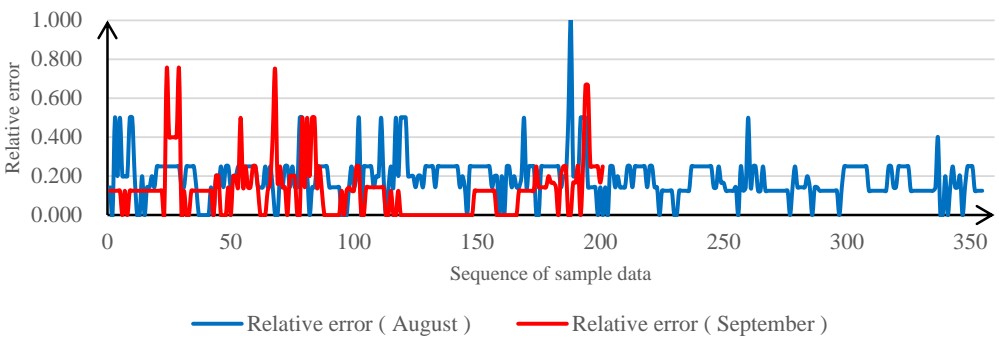

**Figure 6.** The relative error of referred Vis from Equation (4) against observations obtained from the Chinese Ninth Arctic Science Examination (CNASE).

### 3.3. Result of Gridded Vis

In this section, results related to gridded Vis values obtained using the ANN analysis described in the previous section are provided. To test the accuracy of the inferred gridded Vis, gridded Vis data for the Indian Ocean at 00:00 UTC from the 1 January 2010 case is obtained. The parameters of the constructed ANN model utilizing measurements from the January 2010–2017 time period to train the model are first obtained. Then, the gridded data of child nodes are moved to the trained ANN model, and then, Vis at the grid points is inferred. The gridded data of child nodes include the ERA-Interim gridded data and AOD gridded data representing the resolution of 0.125° × 0.125°. The ERA-Interim gridded data set included SLP, Ta, RH, Ts, TDsa, and Uh parameters, as described before.

The statistical result showed that there are 63,206 groups of measured data from 1 January 2010 in the Indian Ocean, but only 52 groups of measured data contained a Vis value at 00:00 UTC. A comparison of ICOADS Vis data to that of the nearest gridded location of the ICOADS data found that the referred Vis is very close to the measured Vis with a relative error of 12% (Figure 7).

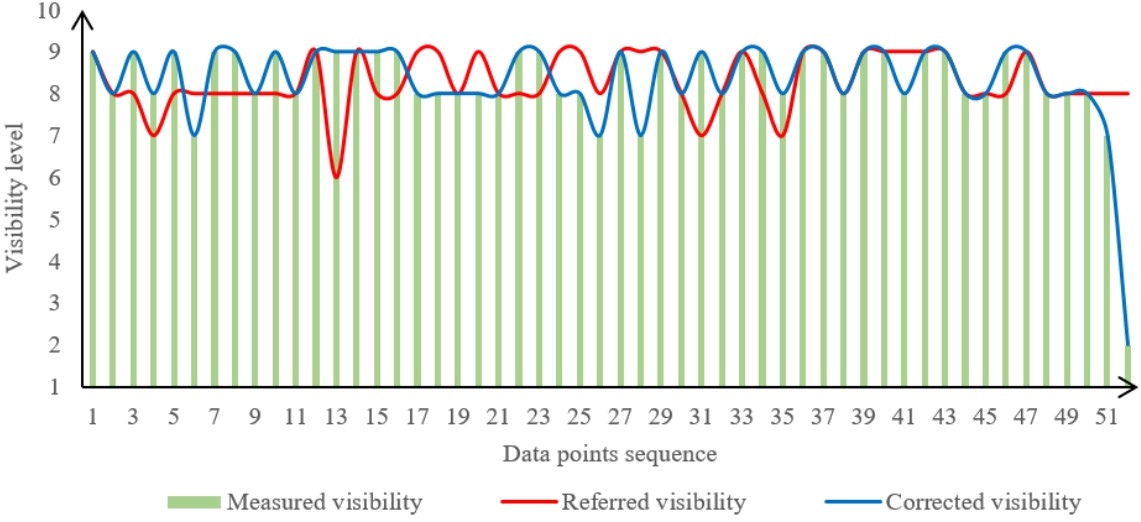

**Figure 7.** Comparison of referred Vis, corrected Vis, and measured Vis.

Since the reanalysis data used to infer gridded Vis have unavoidable errors relative to measured Vis and the parameters of the trained ANN model have unavoidable errors relative to the real condition, it is necessary to correct the inferred Vis using the measured Vis. The inferred Vis are corrected based on the following three steps:

Step 1: Calculate the weight value at the gridded data point representing a 0.125° × 0.125° area. It is assumed that the gridded Vis could be influenced by the measurements within the radius of 14 km around the grid point (because the longest distance between two adjacent grid points over a 0.125° × 0.125° area is less than 14 km). Then, the weight value can be defined as:

$$\lambda_k = \frac{1/D_k(x_i, y_i)}{\sum_{k=0}^{k=Q}[1/D_k(x_i, y_i)]} \tag{5}$$

where $Q$ is the number of measured data points that fall into a circle with a 14-km radius around the grid point $(x_i, y_i)$. $D_k(x_i, y_i)$ is the distance between the kth measured data point at the grid point $(x_i, y_i)$.

Step 2: Calculate the difference between the measured Vis and the referred Vis on the nearest grid point to the measured Vis. It can be defined as:

$$E_k = S_k - P_k \tag{6}$$

where $E_k$ is the difference between the measured Vis and referred Vis on the nearest grid point. $S_k$ is the measured Vis of the kth location, and $P_k$ is the referred Vis on the nearest grid point to $S_k$.

Step 3: The corrected Vis after the above steps is defined as:

$$e(x_i, y_i) = \sum_{k=0}^{k=Q} \lambda_k E_k \tag{7}$$

and:

$$C(x_i, y_i) = e(x_i, y_i) + P(x_i, y_i), \tag{8}$$

where $e(x_i, y_i)$ is the calculated difference between the referred Vis and measured Vis at the grid point $(x_i, y_i)$. The $P(x_i, y_i)$ and $C(x_i, y_i)$ are the referred and corrected Vis values at the grid point $(x_i, y_i)$. It is found out that the corrected Vis is closer to the measured Vis after numerical correction is applied, which greatly improves the accuracy of the inferred Vis (Figure 7).

To further test the accuracy of the referred Vis, the referred Vis for the Indian Ocean case from the 6 to 7 November 2010 case using a spatial resolution of 0.125° × 0.125° is shown in Figure 8 when a tropical storm was crossing the region. The time resolution of the referred Vis is set up as 6 h. The locations marked by green ovals in Figure 8 represent the low Vis centers. The results suggested that the referred Vis accurately reflected the trajectory of tropical storms, which indicated that the inferred Vis is highly reliable compared to the observations.

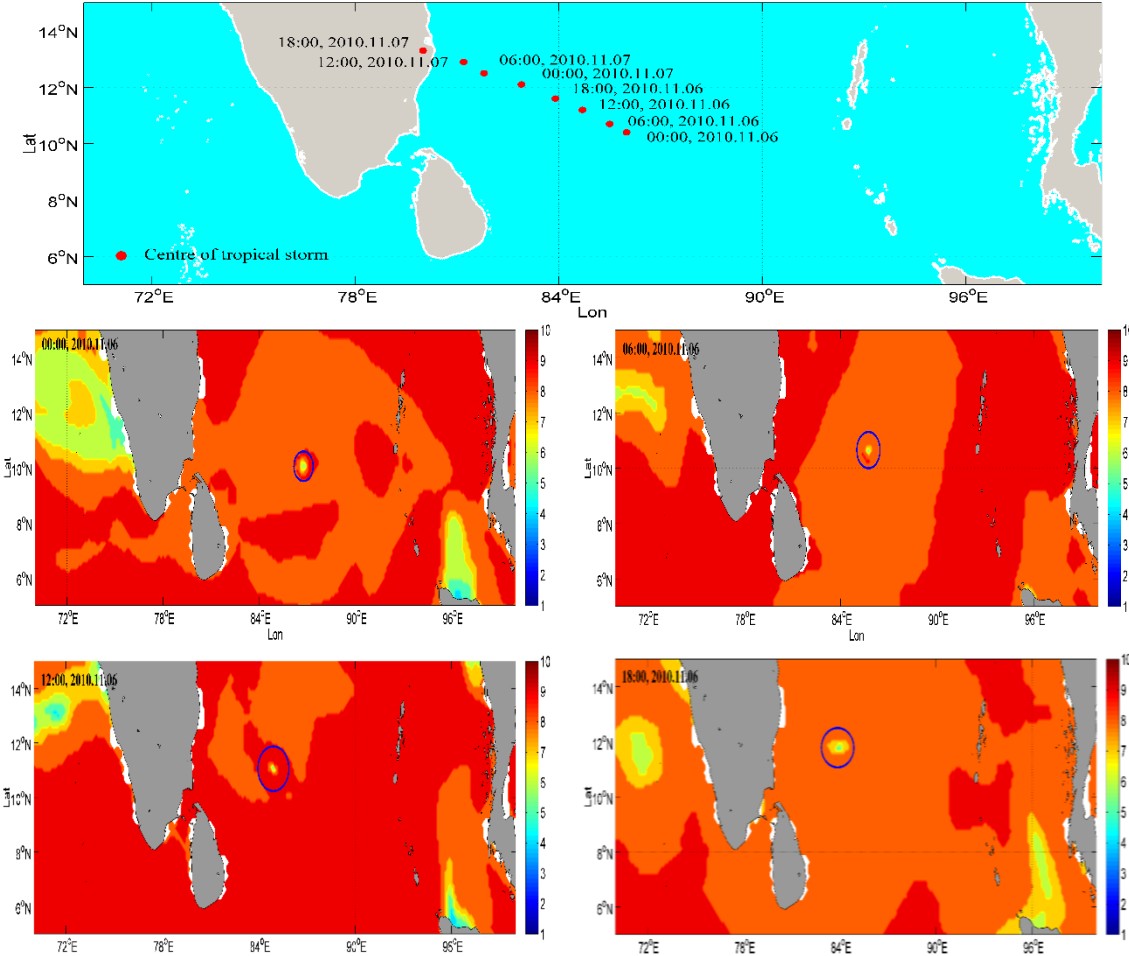

**Figure 8.** *Cont.*

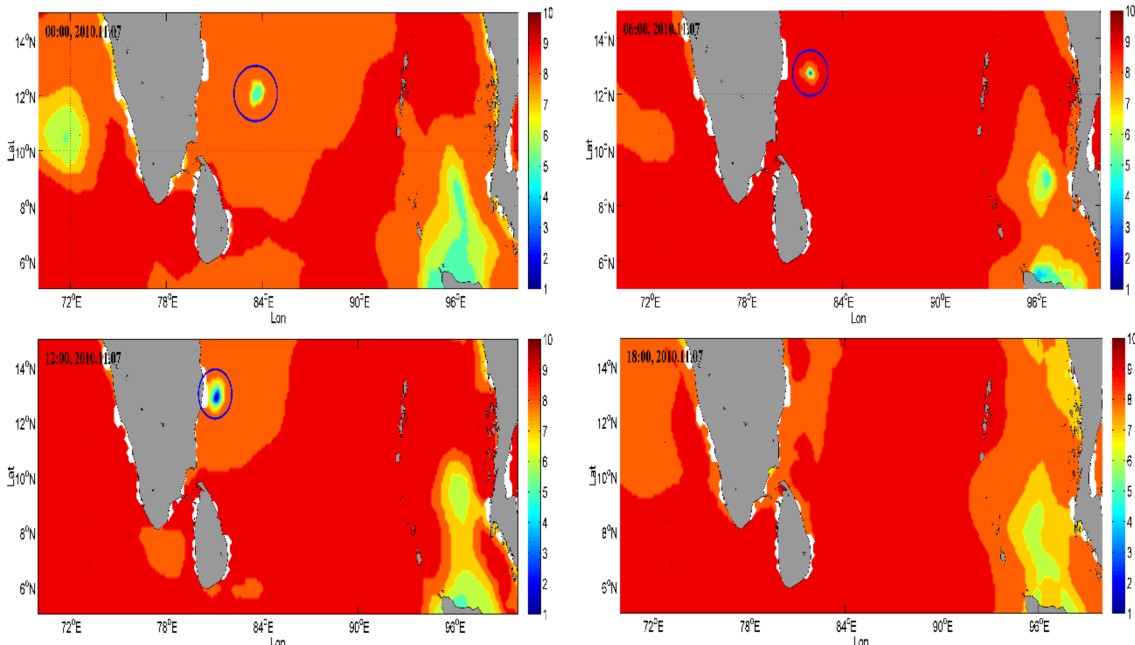

**Figure 8.** Vis results from the ANN method under tropical storm weather conditions. Tracking a tropical storm in the Indian Ocean from 6 to 7 November 2010. Its low-pressure center is indicated by blue ovals. The color bar indicates the Vis level.

## 4. Discussion and Conclusions

Vis representing reflected light intensity seeing by a human has an important impact on the risk analysis of marine navigation systems, but current measurements and the prediction of Vis lack spatial coverage over remote areas such as marine environments and Arctic conditions. Geostationary satellites have global spatial coverage capability, but may not work accurately in northern latitudes and during the daytime because of high-level cloud cover and solar radiation impact. At the same time, Arctic regions still have a severe lack of observational networks [25], and NWPs are limited because of microphysical and boundary layer algorithms [7,8].

This study aimed to discover the feasibility of using a reanalysis of meteorological data from the ECMWF and satellite-based AOD data from the National Aeronautics and Space Administration (NASA) MODIS products to monitor Vis. Certainly, a vertical extinction of visible light from satellites needs to be converted to horizontal Vis and needs geostationary polar orbiting satellite data in northern latitudes. The results showed that ANN analysis for Vis monitoring can promise better results representing remote marine environment conditions than conventional methods.

Although the inferred Vis has a higher accuracy than that of NWP and PCR predictions, an important issue still exists in the current analysis. Figure 8 suggests that inferred Vis values had NaN values near shorelines because Ts is obtained from ECMWF analysis with NaN values, and this resulted in bad values in inferred Vis. This shows that the uncertainty still exists in the inferred Vis due to the uncertainty found in the gridded data of child nodes.

Reducing the uncertainty in the analysis requires improvements in the quality of collected data and microphysical algorithms in the NWP model simulations. The microphysical model of atmospheric Vis needs to be significantly improved, especially for sea fog [4]. At the same time, with the development of Big Data technology, a deeper network could be used to infer Vis to reduce the uncertainty of the inferred Vis in the future. Meanwhile, improving the model by trying to take other factors influencing Vis into consideration is also necessary to reduce the uncertainty in the future. Since the method used to calculate AOD is the Inverse Distance Weighting Method, which is a common and simple method, better methods to calculate AOD will be studied in the future.

The current work suggested that Vis has a very important effect on the sea transportation safety, likely reducing accidents; therefore, a large amount of high-quality Vis data should be the basis for evaluating the navigability risk over remote areas. At the same time, the generated large amount of high-quality Vis data could be input into the Vis prediction model as an initial field to predict Vis in the future, which is so scarce now. The method used to generate gridded Vis in the paper can also be used to directly predict Vis in the future. If the objective is to predict Vis in a specific location, the constructed model should be trained with the data available for that location.

The overall results suggested that improved Vis products can be obtained using ANN analysis, but Vis prediction in foggy environments is still an issue because of the uncertainty exists in microphysical parameters simulated by the numerical models. This needs to be further studied. At the same time, large data sets should be used to adequately train the parameters of ANN. Perhaps, a better designed field project can help to develop improved ANN analysis for Vis predictions for marine environments.

**Supplementary Materials:** The following are available online at http://www.mdpi.com/2076-3417/9/21/4487/s1.

**Author Contributions:** Y.S. is the first author of this paper and the experiments and writing are mainly done by him. R.Z. is the first corresponding author of the paper and he provided the ideas for writing the paper. I.G. is the second corresponding author of the paper and he helped improve the experiments and writing of the paper. Y.Z., M.L. and Y.W. helped collect data and revise the paper.

**Funding:** This research was funded by National Natural Science Foundation of China (grant number: 41976188) and the APC was funded by National Natural Science Foundation of China (grant number: 41976188).

**Acknowledgments:** Measured data used to train the model were distributed by the NCDC (National Climate Data Centre of the United States). AOD was obtained from the MODIS Level 3 AOD product, which was published by NASA. Reanalyzed gridded data that were used to infer the gridded visibility were obtained from the ECMWF. Data of tropical cyclones were used to further test the accuracy of inferred Vis is from the JTWC.

**Conflicts of Interest:** The authors declare no conflict of interest.

## Abbreviations

| Acronyms or Symbols | The Definition of the Acronyms or Symbols Used in the Text |
| --- | --- |
| ANN | Artificial neural network |
| $aod_k$ | The AOD at the location k |
| AOD | Aerosol optical depth |
| BP | Back propagation |
| CCN | Cloud condensation nuclei |
| $C(x_i, y_i)$ | The corrected visibility at the grid point $(x_i, y_i)$ |
| $D_k$ | The distance between the location k and the position of ICOADS data located |
| ECMWF | European Centre for Medium-Range Weather Forecasts |
| $E_k$ | The difference between measured visibility and the referred visibility on the nearest grid point to the measured visibility |
| $e(x_i, y_i)$ | The calculated difference between referred visibility and measured visibility at the grid point $(x_i, y_i)$ |
| ICOADS | International Comprehensive Ocean-Atmosphere Data Set |
| JTWC | Joint Typhoon Warning Centre |
| LWC | Liquid water content |
| MODIS | Moderate Resolution Imaging Spectrometer |
| Nd | Droplet number concentration |
| NWP | Numerical weather prediction |
| NMPF | Microphysical parameterization for fog |
| NASA | National Aeronautics and Space Administration |
| NCDC | National Climate Data Centre of the United States |
| PCR | Principal component regression |
| $P(x_i, y_i)$ | Referred visibility at the grid point $(x_i, y_i)$ |
| $P_k$ | The referred visibility on the nearest grid point to $S_k$ |

| Acronyms or Symbols | The Definition of the Acronyms or Symbols Used in the Text |
|---|---|
| RH | Relative humidity |
| RMSE | Root mean square error |
| SLP | Sea level pressure |
| $S_k$ | The measured visibility of the kth location |
| Ts | Sea surface temperature |
| Ta | Air temperature |
| TDsa | Ts-Ta |
| Uh | Horizontal wind speed |
| Vis | Visibility |
| $V_c$ | The referred visibility |
| $V_r$ | The measured visibility |
| $\lambda_k$ | The weight of the location k |
| $\delta$ | Relative error |

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
