# Peer review of "Gridded Visibility Products over Marine Environments Based on Artificial Neural Network Analysis"

_applsci, doi:10.3390/app9214487_

Round 1

Reviewer 1 Report

Τhis is a very important work that would be useful to the scientific community and the general audience. Visibility is an important variable for many applications. The text is more focused on marine transport but to my knowledge it is more important for aviation (for landing/taking off)  and I suggest to add some discussion on that. Also the method developed in this study looks very promising and would be great to move to operational status. Thus, I suggest publishing the paper after some revisions in some parts that are not so clear.

Abbreviations should be defined in the main text. Thus, defining them in the abstract is not enough. Vis and NWP is nowhere defined in the main text.

The introduction part needs some rewriting. I think it is important to add some information on the methods used for in situ and remote sensing retrieval of visibility. Also, parameters affecting visibility as a physical/optical process should be discussed in a more general way, before refereeing to studies and methods already in use. It is difficult for the reader to evaluate these info without having an image of the science implied by it.

P.2.L40-42 some discussion about recent works that sudy the role of pollution to visibility should be added (eg Founda et al., ACP 2016, Singh et al. , ACP 2017).

P3.L1 all these variables are not obvious. They should be defined.

P3 L10 reliable historical gridded data. I assume you mean that the test data are in the past, but I think “historical” is a misleading term here.

P3 l15-21 These scheme description is too short and is difficult to be evaluated. I suggest to expand it with more details. Which meteorological parameters where studied first, how the relationships where set up. How the factors where classified. Which parameters were eliminated.

P4 l4-6 The sentence should be rewritten. It seems that AOD are from ECMWF, and In that case it would be no grid position problem.

P4 The method here is an independent model to forecast AOD. This should be stated more clear. Because the uncertainty of this forecast would propagate to the final product.

P5 l3-4. This sentence looks more like an advertisement than a scientific paper. Please elimante.

P5 l28-31 Ideal values where never defined clearly. Define earlier in the next what you are looking for in the output.

P5l34-36 Explain how these values where selected.

P6l1 you claim that 30 experiments were repeated  but at l9 you state 10 experiments. Please correct.

P6l 17-19 This is true if the clouds are low. If the clouds are high, then aerosols will play an important role in visibility. I suggest to discuss this effect more clearly

Visibility level should be defined more clearly , because it is important to understand the figures and the table is not helpful.

P7l7-p8l1 This sentence is very badly written and does not offer anything. I suggest skipping it.

P8 l2-4 This is probably the most important finding of the study. That including AOD in the calculations makes very slight difference to the result. I could not understand it as a physical process, but I assume somehow there is a proxy connection in the neural network. Thus, I think it is critical to discuss this finding in detail and investigate more. Are the data with high differences of relative error (blue and yellow at fig 5) related to high AOD values? Could you plot a scatter of relative error (AOD)-relative error(no AOD) in respect to AOD, and probably export a relation? Or set a threshold of when AOD becomes important for the model?

P8l26-p9l2. Are the meteorological /marine data anywhere used in the study? I guess no. In that case I don’t see the point to mention them

P10 ICOADS vis data are never explained, what type of measurements provided this data set? Observational? Remote sensing? What’s the expected uncertainty?

Section 3.3 I think that the rationale of this section is not clear. A method to corrected the output is presented, but is not clear how it comes the need for that.

Figure 9. I guess the colorbar is visibility level. Is nowhere explained. Also, is it measured or output of the model? How to validate /evaluate this result? It seems a showcase of the output but without anything to compare.

P12 25. Something is missing in this sentence

Conclusions I suggest discussing more the important findings of the study regarding AOD

Reviewer 2 Report

The study aims to improve Vis prediction and reduce navigation risks by using a gridded Vis dataset and an artificial neural network.

In order to fit better with the scope of the special issue the relationship with the problem of air quality prediction should more a bit more developed (especially in the introduction). Also, in the discussion, the authors should explain if and/or how their method could be applied for the prediction of air contamination in a more general context (e.g., urban air pollution).

As minor observation, an extra “that” must be removed at the end of line 36.

Author Response

This manuscript is a resubmission of an earlier submission. The following is a list of the peer review reports and author responses from that submission.

Round 1

Reviewer 1 Report

Overall, my opinion is that the scientific quality of the work is inadequate to warrant a publication. More specifically, there are some gaps in the proposed methodology for visibility inference that raise concerns. Vis in areas with meager meteorological ground observations is a legitimate problem. I would say this is more of a problem for coastal navigation and not a problem for open sea navigation because RADAR solves this problem completely. In trafficked seaways, visibility observations are available and at any rate a RADAR working at all times solves the problem. So the claim that poor visibility is a hazard that mariners need to know in advance is not correct. No ship will alter course to avoid an area of low visibility. The initial claims of the authors to justify the use the work indicate that they have no knowledge of modern marine operations. Also, turbulence is not a navigational hazard.

Let’s talk about the science. First, point of concern is the radius R of the extrapolation of AOD. A reference is made to a statistical model governing this extrapolation, but the model is never mentioned. Instead, formulas 1,2 indicate that AOD extrapolation is made using a convex combination of measured AOD, but this formula is ad hoc, reasonable, yet not justified. Such a simplistic model can be relatively accurate only for small R, and this explains to me the relative error being the smallest for R=20km. Fig. 3 should have started with this R value and not from 10km because your input data are gridded for 10x10 km^2 resolution, so no ground truth exists in between the vertices of the grid. I think this could have been better done with averaging over several iterations of simulated annealing and for a radius no greater than 100km, with existing AOD values serving as boundary value data. In the same spirit, I have concerns for the extrapolation of visibility using the simplistic neural network. I think a deeper network which inputs also existing Vis values and AOD and other measurements would have worked   better., because this is the objective of the work.

One last point; the validation of the AOD extrapolation is vaguely described. You split the data in two parts. How many times do you repeat this evaluation process? It has to be done several times and report the average values of error over all repetitions, not just the average relative error as in Fig. 3. Overall, the work from the viewpoint of data analysis is superficial and the methods not well thought.

Reviewer 2 Report

Dear authors,

The paper quite interesting as a whole but need major review. The major points should be addressed:

Define Nomenclature, Greek symbols, subscripts, superscripts, acronyms and abbreviations separately in table form. Please provide analysis for RMSE or NRMSE between estimated and source data. Relative error is insufficient. Please add the appropriate charts. Please specify precisely how the input data to the model has been aggregated, Please add the scheme with the adopted and described structure of the neural network, For training the ANN, which method was used? Levenberg Marquart or Besian Regulation or Scale conjugate? Please add a paragraph about training the algorithm. Please add the performance output plot. Where is MSE (mean square error) plot? Where is error histogram? Where is regression plot? Page 6 line 9-12: ,,one of the reasons is the 10 inadequate training of ANN due to incomplete data set; the second reason is the deficiency of the 11 model, such as the selected factors influencing Vis are not complete. The last reason is the inaccuracy 12 of the data set used to test the accuracy of the trained model”. Is it possible to eliminate these restrictions to increase accuracy? Relative error greater than 2-4% gives unsatisfactory results. In the summary, please add information about future work.

Reviewer 3 Report

The authors present an ANN-based model to predict a visibility indicator over marine environments using atmospheric analysis.  

Strengths: 

The problem statement is well explained, and the scientific merit of the question is clear.  The methodology is not specific to geographic features, allowing it to theoretically generalize to any remote region. 

Weaknesses:

The reader is sent to another paper for any model detail.  This is unacceptable in a paper exhorting the value of a model.  What little detail there is in unclear (e.g., is the model 3,5, or 7 layers?  There is textual evidence for all 3). How are hyperparameters optimized in this case?  How does this effect generalizability?  The generalizability question is especially worrisome in light of the "tweaks" needed to obtain low errors.  I do not know enough about this domain to know how self-similar your applications areas are, but as is I'm not convinced that this isn't overly engineered to a given scenario (where you have a ground truth to continuously check against) Finally, there is a claim that ANN is superior to other methods, but there is no parallel presentation of Ann results against those of conventional methods.  Given that, it is difficult to understand this claim. 

Round 2

Reviewer 1 Report

First, I find responses 1-4 offensive. Turbulence at sea, where does it exist in the open ocean? Tell me one incident. When I spoke about the use of RADAR I meant its use to navigate not to predict precipitation or fog. Poor visibility may increase the chance of an accident, but, my experience with marine operations taught me that proper speed and use of RADAR minimize the chances of error. This might not be the case in China straits. Seamanship and RADAR use prevent collisions not meteorological predictions. So the comments added by the authors in the introduction to justify the significance of the work in reference to mariners are not valid and should be removed. I think that the prediction of visibility is a nice problem with its own merits in the science of meteorology. Speaking of science, I see that no significant changes have been made to address my previous concerns. First, let's simplify the problem. Since you have referenced visibility why not construct as simple model to infer visibility from the recorded data and include some short time prediction, perhaps. 

You basically, with the last algorithm duplicate your inference for AOD using a convex combination of inverse distances from data points with ground truth. First, you say you use a statistical model referenced in your paper, but this is a model applied for a part of Taiwan and for rain. Why should it apply over oceans and for AOD. There is no reason. In short, you never addressed this concern from my previous report. Apart from that you also report in your data comparison analysis a small error between the model using AOD and not using AOD. Since, Vis is graded with integer values a reasonable error is +-1. This is shown in your data and had you taken our the AOD you would see that your results don't change much. 

On the other hand, I look at your validation results. They use the month of January only. What happens with other months? This is a serious issue because it concerns the reproducibility of the study. Overall, I regret to say that I find your analysis not sufficient to warrant a publication. Finally, your addition on multilayer layer NN is not necessary for the content of the methods section. You just use a simple off-the-self Matlab NN. No need to waste a page on that. It would have been more useful to see your pseudo algorithm for the interpolation of the AOD and calculation of R. BTW, the AOD prediction problem looks to me more like an optimization problem where we have some data and we want to infer some other missing in a way that we do not distort the known data. 

Figures 5 are confusing. There are two vertical axes. Only the left is enough because the reasonable error is +-1 and in most cases, this is what you have. In Fig. 6 your errors are more but in most evaluated case the measured visibility is high. You have no reported data in this figure with low visibility. This is bothersome. Also in the tropical storm, you present the calculated visibility in the area close to the center of the storm to be high (not the eye), although this is counterintuitive due to rain notwithstanding the aerosols of sea droplets blown by the prevailing winds. Too many questions, my recommendation is to withdraw it and rework it and ask for some professional data analyst to help you.

Reviewer 2 Report

Authors must take all suggested changes from the original review.
Not all corrections have been made in the current form of the manuscript.

The manuscript must be supplemented and corrected in detail.